# The Effects of Brackish Irrigation on Soil Ion Accumulation and Growth of *Atriplex* Species

Sarah M. Cerra [1,*], Manoj K. Shukla [1,*], Soyoung Jeon [2] and Scott O'Meara [3]

[1] Department of Plant and Environmental Sciences, New Mexico State University, Las Cruces, NM 88003, USA
[2] Department of Economics and International Business, New Mexico State University, Las Cruces, NM 88003, USA
[3] Bureau of Reclamation, Denver, CO 80225, USA; someara@usbr.gov
[*] Correspondence: scerra@nmsu.edu (S.M.C.); shuklamk@nmsu.edu (M.K.S.)

**Abstract:** Prolonged drought conditions in New Mexico have led growers to use brackish groundwater for crop irrigation. Desalination of the groundwater with reverse osmosis (RO) is possible, but the concentrated waste requires environmentally safe disposal, such as by irrigating native halophytic plants, *Atriplex*, which could be cultivated to feed livestock. We hypothesized that ions from the brackish irrigation would increasingly accumulate in the soil away from the roots as the wetting front expanded further from the emitter, while not affecting the aboveground growth of the plants. *Atriplex* species were irrigated with brackish water at two irrigation levels for three years. Soil samples were collected at the beginning, middle, and end of the study at two depths and three distances from the emitter. Electrical conductivity (EC), soil ion accumulation, and plant growth were recorded. The average EC of the soil increased with brackish water irrigation. As the ions accumulated along the wetting front of the percolating water rather than near roots, a favorable environment for root growth was provided. While sodic levels of ion accumulation were not reached in this study, aboveground growth still declined. This leads to the recommendation that RO-concentrated waste could be used to irrigate *Atriplex* species for livestock fodder, with further plans to irrigate with fresh water to remove accumulated ions as a potential sustainable waste management process. Additional studies are necessary to develop guidelines for *Atriplex* tolerance and harvesting.

**Keywords:** RO-concentrated waste; soil salinity; *Atriplex* species; electrical conductivity; drip irrigation

## 1. Introduction

The western United States have experienced higher than average temperatures since 2000 [1], with some areas increasing by approximately 1.5 °C between 2000 and 2018. This increase in average temperature is associated with a decrease in precipitation [1]. As availability of surface water for irrigation in New Mexico decreases [2,3], growers need a supplemental water source for crops that require fresh water. In New Mexico, 75% of the groundwater is brackish (electrical conductivity (EC) > 3 dS/m) [4], and desalination of groundwater is necessary prior to crop irrigation or human consumption [5].

From 2010 to 2020, global freshwater production through desalination processes increased by 6.8% [6]. Reverse osmosis (RO) accounts for 85% of the operational desalination plants in the world [6]; however, costs associated with energy are high [7], and concentrated waste management is a challenge, specifically for inland desalination plants [8]. Traditional disposal methods of concentrated waste include discharge to surface waters and oceans through deep well injection and thermal evaporation, which may all have ecological costs [9]. Other concentrated waste-disposal methods, such as zero liquid discharge technologies and the use of waste as an irrigation source, are still being developed [8,9]. It was hypothesized that there would be minimal effects of the concentrated waste when it was applied to native halophytes [9–11]. In addition to understanding the effects of

RO-concentrated waste application on the halophytes, it is also important to understand the effects of the concentration's application on the soil and the plant root zone.

Soil salinity is of great concern globally as semi-arid and arid lands increase due to changes in the climate [12]. Soil is described as saline if there are excess levels of soluble salts (EC > 4 dS/m) in the soil water, and it is described as sodic if the SAR (sodium absorption ratio) is 13 or greater [13]. Because the soils of southern New Mexico typically have an alkaline pH and an EC below 4 dS/m, research [14] has suggested that soil sodicity be considered a problem needing attention when irrigating with brackish water. A SAR of greater than 13 will reduce the nutrient availability and hydraulic conductivity of the soil [14]. Sodic soil degrades the soil structure as clay particles and sodium ions fill pores, reducing water flow through the soil [13,15].

Delivery methods of irrigation must be considered in order to ensure water conservation in arid and semi-arid regions. Flood irrigation is frequently used in the Rio Grande Valley for agriculture [16]. This form of irrigation allows for ions to be transported through the root zone, allowing ions to accumulate in lower soils and groundwater aquifers as irrigation water moves through the soil [17]. Drip irrigation has been used steadily in many arid and semi-arid regions to reduce evaporation and percolation to ensure water conservation [18,19]. Subsurface drip irrigation used in Israel in olive orchards showed an increase in soil salinity above the emitter [18]. Generalized patterns of ion accumulation were difficult to predict in the subsurface irrigation; however, one study [18] was able to characterize more accumulation of ions between emitters where wetting fronts over lapped, and less accumulation near olive trees, where roots could be taking up ions. When modeling drip irrigation wetting patterns for sandy loam, researchers were able to determine a radius and depth of wetting [19].

Many crop plants are stressed by both osmotic potential and biochemical process reductions, which reduces yield when introduced to saline and sodic soils [20]. However, in halophytic plants, toxic concentrations of ions leading to salinity do not appear to be reached [8,21]. Halophytes have been noted to compartmentalize ion accumulation, adjust the osmotic potential, and secrete salts from specialized glands [22–25]. Because halophytic plants are not affected by salinity stress at the same levels as other species, these plants could be irrigated with RO-concentrated waste. While many halophytic plants can uptake salts from the soil, *Atriplex* has a large amount of biomass when compared to other halophytes, making it the better candidate for land management practices to remove salts from the soils [26]. Studies [22,27] found that biomass did not decrease when RO-concentrated waste was used on the halophytic species studied when irrigated for one year. When soil salinities rose to near sodic levels, some halophytic were able to compensate for the salts [27,28]. The authors reported a reduction of 38.5% in soil salinity and 33% in soil sodicity in the top 10 cm of soil when *Atriplex* species were planted for soil remediation in [25]. A second study [29] also reported decreases in soil salinity and soil sodicity 18 months after planting *Atriplex nummularia*. How well *Atriplex lentiformis* and *A. canescens*, native species to southern New Mexico, are able to uptake soil ions to remediate the soils has not yet been studied.

Many of the studies examining the effects of irrigation using brackish or saline water waste on agricultural plants do not address the long-term effects of irrigation on ion accumulation in the soils. This study aims to address the impacts of ion accumulation in soils for three years of drip irrigation with saline-concentrated waste on *Atriplex canescens* and *A. lentiformis* to simulate reverse osmosis-concentrated waste effects. The objectives of the study are to (i) describe the pattern of electrical conductivity (EC) to indicate soil salinity increases; (ii) describe individual ion accumulation in the soil for differing irrigation regimes over a three-year period; and (iii) compare irrigation treatment effects on the aboveground growth of *Atriplex* species over three years. We hypothesized that electrical conductivity would increase in an inverted bell pattern as ions from the brackish irrigation increasingly accumulate in the soil away from the roots as the wetting front expanded further from the emitter over a three-year period. Given the halophytic nature of *Atriplex*, we did not expect

the aboveground growth of the plants to decrease over three years with the increased soil salinity, regardless of irrigation treatment. Understanding how the ions accumulate as concentrated waste is applied to the soil through drip irrigation will help growers and land managers determine steps to ensure that the root and plant growth in saline soils is favorable. Ensuring favorable root growth environment is necessary for the growth of the plant and the potential use of the plant as fodder when attempting to remediate soils of accumulated ions.

## 2. Materials and Methods

### 2.1. Site Description

The field study was conducted at the Brackish Groundwater National Desalination Research Facility (BGNDRF) in Alamogordo, New Mexico (32°52′ N, 105°58′ W, elevation 1322 m). The area receives, on average, 33.6 cm of annual precipitation (Supplemental Figure S1) [1]. The yearly average maximum temperature is 23.89 °C, and the annual average minimum is 8.33 °C (Supplemental Figure S1) [1]. The region is classified as a semi-arid desert.

Two half-acre plots within the agriculture research area (total 0.4 ha) at the Brackish Groundwater National Desalination Research Facility (BGNDRF, Bureau of Reclamation) were identified (Supplemental Figure S2). Both plots drain from the northeast to the southwest, although low points within each field allow for occasional ponding. Agriculture plots were planted between April 2014 and October 2015 with *Atriplex* at a rate of 1 plant per 2 m, in 6 zones. Zones were identified as 1–6 from west to east in each plot. The drip irrigation line of Netafim Uniram Heavywall was placed on the west side of each row of plants, as close to the base of the plant as possible. Irrigation lines were approximately 3 m apart. Emitters within each line were placed at 0.3 m intervals. Each zone contained two rows of plants of a single species, oriented in a north to south fashion. Each zone was randomly assigned an irrigation rate.

The research facility includes four groundwater wells, varying in salinity (Supplemental Figure S2A). In February 2017, all zones were leached with water from well 1 (Total Dissolved Solids of 1240 mg/L) [30]. Leaching occurred through drip irrigation three times per week for 120 min during each irrigation, amounting to 9.4 L per plant per irrigation event. This continued for 6 weeks to remove previous ions accumulated in the soil. In April 2017, following the leaching, soil samples were collected to establish baseline data. All plants in the plots were trimmed to 95 cm in height and 60 cm in diameter for uniformity. Irrigation treatments began immediately after baseline soil samples collection.

### 2.2. Soil Sampling

Six plants total were randomly selected as study plants before irrigation treatments were randomly assigned to each zone, three replicates for each irrigation treatment. Plants were selected away from the edges of the plots. Test plants were selected via visual appearance of health, including robust leaf appearance with no obvious signs of distress. Selected test plants were also observed not to have differences in the soil topography (dips or hills). Two irrigation treatments were assigned randomly to the test plants. At each plant, three distances from the irrigation were selected for soil sampling, duplicated on either side of the irrigation emitter, east and west (Supplemental Figure S2B). These distances included 30 cm, 60 cm, and 90 cm. Sampling of soil included two depths, 0–25 cm and 25–50 cm (Table 1). A total of 12 soil samples of approximately 500 g were collected at each of the six test plants three times over the course of the study (Table 1). These samples were collected in April 2017 to determine a baseline at the end of the first growing season in November 2017 and at the end of the third growing season in November 2019.

**Table 1.** The study parameters included six locations for soil sampling. The circumference of each plant in the study was also measured before trimming back yearly growth to determine aboveground growth.

| Irrigation (3 Replicates) | Distance from Emitter (2 Sides) | Soil Depth | Time of Sampling |
|:---:|:---:|:---:|:---:|
| 80% ET0 | 30 cm | 0–25 cm | Baseline |
| 60% ET0 | 60 cm | 25–50 cm | End of Year One |
| | 90 cm | | End of Year Three |

### 2.3. Irrigation Scheduling

Irrigation treatments were established using the Blaney–Criddle equation rather than the Penman-Monteith method due to missing weather data necessary to complete the PM method. The Blaney–Criddle equation is as stated [31]:

$$ET_0 = p(0.46T_{mean} + 8),$$

where p is the percentage of sun for latitude of 35° N, and $T_{mean}$ is the average monthly temperature in degrees Fahrenheit. The resulting ET0 (reference evapotranspiration) was converted from inches to millimeters. We calculated the ET0 to be 1335 mm/year for 2017. At this ET0, 6635.9 L of water would need to be applied to each plot weekly. It was then determined that because in a natural setting with water availability decreasing, irrigation treatments were set at 60% and 80% of the use of water established, meaning we would need 5307 L/week and 5686 L/week, respectively.

Irrigation events occurred three times per week with a mixture of Well 2 and Well 3 at BGNDRF (Supplement Figure S2) to create a water solution of approximately 4200 mg/L TDS (Table 2) [30]. This mixture at the rates applied mimics the effects of using RO-concentrated waste. RO-concentrated waste could not be directly applied to the field due to regulations in the United States; permits allow the salinity of irrigation to be no more than the salinity of the ground water to prevent groundwater contamination. Irrigation events lasted for 88 min to represent the 60% ET0 and 120 min to represent the 80% ET0. Irrigation events were not adjusted for a changing monthly ET0; however, irrigation was paused from November to January in each year of the study. Irrigation was suspended one week prior to soil collections and plant trimmings.

**Table 2.** Average ion concentrations (in mg/L) of Well 2 and Well 3 at BGNDRF through the study period. Full well data are available online [30].

| Parameter Name | Reporting Units | Well 2 | Well 3 |
|:---:|:---:|:---:|:---:|
| Total Alkalinity (as $CaCO_3$) | mg/L $CaCO_3$ | 210.17 | 181.00 |
| Chloride | mg/L | 582.83 | 649.67 |
| Hardness, Total (as $CaCO_3$) | mg/L | 2463.33 | 1830.00 |
| Nitrogen, Nitrate (as N) | mg/L | 6.63 | 2.55 |
| Calcium | mg/L | 489.67 | 424.33 |
| Magnesium | mg/L | 301.17 | 188.33 |
| Sodium | mg/L | 652.5 | 350.63 |
| Potassium | mg/L | 2.44 | 2.93 |
| Solids, Filterable Total Dissolved Solids | mg/L | 5155.00 | 3420.00 |
| EC (calculated from TDS) | dS/m | 6.45 | 5.34 |
| pH—Aqueous | pH units | 7.35 | 7.35 |

Plants were trimmed to a set height of 95 cm and a diameter of 60 cm prior to each growing season (2017, 2018, and 2019). Before trimming the plants, the circumference of the six studied plants was recorded as aboveground biomass. These data were analyzed to determine differences in growth of aboveground biomass.

### 2.4. Physical Analysis

Soil texture was identified from 16 stratified random soil samples in April 2017. Soil was air dried and passed through a 2 mm sieve. The hydrometer method [32] was used to determine sand, silt, and clay content. Soil texture was determined using USDA (US Department of Agriculture) texture triangle.

### 2.5. Chemical Analysis

Each soil sample was air-dried, and large organic material was removed as 100 g of soil was placed into a cup. Deionized water was mixed into the soil, creating a paste, at a ratio of 2:1 deionized water to soil. The soil paste was covered with plastic wrap and allowed to rest for 2 h due to time limitations in the lab. Soil paste was then transferred to a Buchner funnel with a 100 mm diameter filter. A 50 mL tube was attached to the funnel, vacuum suction was applied, and effluent was collected. Electrical conductivity (EC) and pH were measured from the effluent [33]. If the EC was greater than 15 dS/m, the effluent was diluted by 10% to obtain a more accurate reading. This testing occurred in April 2017 (baseline), November 2017 (end of year 1 or EOY 1), and December 2019 (end of year 3 or EOY 3).

Total magnesium, calcium, sodium, potassium, nitrogen (as nitrate), and chloride ions were measured from the effluent through ICP ion analysis [34]. The sodium absorption ratio (SAR) was calculated according to the following equation [35]:

$$\text{SAR} = \frac{[\text{Na}^+]}{\sqrt{\frac{([\text{Ca}^{2+}][\text{Mg}^{2+}])}{2}}},$$

where $\text{Na}^+$ is the concentration of sodium ions (as meq/L), $\text{Ca}^{2+}$ is the concentration of calcium ions (as meq/L), and $\text{Mg}^{2+}$ is the concentration of magnesium ions (as meq/L). Soil chemistry analysis occurred at the New Mexico State University soils lab for the baseline data. Approximately 100 g of soil from each sample was sent to AgSource Laboratories (Lincoln, NE, USA) for soil chemistry analysis for 2017 and 2019.

### 2.6. Statistical Analysis

Two irrigation treatments were organized in a completely randomized scheme of three replicates each, assigned to one of the test plants each. Differences for each distance from emitter treatment and soil depth treatment compared to time (Baseline, End of Year 1, and End of Year 3) were determined using analysis of variance (ANOVA) tests. Tukey's multiple tests were performed for between-group comparisons. Data were standardized to conduct principal component analysis. Statistical analysis was conducted using R (version 4.0.2), and $p$-values less than 0.05 were considered statistically significant.

## 3. Results

We found that as the wetting front of water moving through the soil expanded away from the emitter, the accumulation of ions increased, causing the electrical conductivity to be similar to an inverted bell pattern. This was an expected result. Ion accumulation in the soil varied among the different ions. The brackish irrigation caused the aboveground growth of the *Atriplex* species to decrease over three years, contrary to our hypothesis. *Atriplex* grown with 80% ET0 irrigation had more growth than those grown with 60% ET0 irrigation.

### 3.1. Physical and Chemical Analysis

The soil texture ranged from sandy clay loam to clay loam in the 0–25 cm depth and from sandy loam to clay in the 25–50 cm depth (Supplemental Table S1).

The soil EC showed significant differences according to location ($p < 0.001$), time ($p = 0.001$), and irrigation ($p = 0.014$) treatments (Table 3). However, differences in depth were not significant ($p = 0.787$). The EC of combined soil samples 30 cm from the emitters (average EC of 18.66 dS/m) were significantly lower than the EC of soil samples 60 cm (31.88 dS/m) and 90 cm (20.43 dS/m) from emitters ($p < 0.001$). At the end of year 3 (EOY3), soil EC was significantly higher than baseline (BL) and the end of year 1 (EOY1) EC ($p = 0.004$ and $0.007$, respectively), showing an accumulation of salts from an average of 20.10 dS/m in BL samples to 30.12 dS/m by EOY3. Accumulation was approximately 5 dS/m greater in the 60% irrigation treatment in comparison to the 80% irrigation treatment. A significant interaction between location and irrigation was also found ($p = 0.011$) (Figure 1A). The spread of data included more variation from the mean at the 90 cm and 60 cm west sampling points compared to the central sampling points for both irrigation rates (Figure 1B). In general, the lower irrigation rate of 60% ET0 was more variable regardless of the sampling location. Most EC values were clustered below 50 dS/m. Ion accumulation compared to EC values found clusters by ion type with calcium to have the highest concentrations, while nitrate had the lowest (Supplemental Figure S3). The values of EC in the study area increased with ion accumulations; however, accumulations of each ion varied spatially (Supplemental Figure S3).

EC increased as distance from the drip irrigation line increased throughout the study. The 60% ET0 irrigation treatment most clearly showed the expected inverted bell shape (Figure 2B). In the 80% ET0 irrigation, the wetting expanded past 90 cm; therefore, the bell shape was not as clearly defined. At 90 cm from the dripline, at 0–25 cm depth, the EC of these samples was also the highest, at nearly 35 dS/m, and nearly all ions were at the highest levels in the samples taken from this location.

Sodium absorption ratios (SAR) are significantly different (when compared to the irrigation rates) among the baseline, the end of year 1, and end of year 3 ($p = 0.001$) (Supplemental Figure S4). The baseline samples had an average SAR of 10.89, and those at distances greater than 30 cm from the plant were near or above sodic rates of 13 ($p < 0.001$) [36,37].

Although we did not collect an exchangeable sodium, we did calculate the exchangeable sodium percentage (ESP) from the SAR. Statistically, this value followed the same significance pattern as SAR.

### 3.2. Ion Concentrations

The ions studied were calcium, magnesium, sodium, potassium, nitrogen (as nitrate), and chloride. The accumulation patterns of the ions spatially varied (Figure 2). Calcium ions accumulated equally across each distance. Nitrogen concentration appeared to be the ion closest to following the inverted bell pattern from the irrigation emitter. Other ions studied followed a similar inverted bell pattern with greater accumulations west of the emitter. This is further discussed in the next section.

### 3.3. Distance from Emitter

Statistical differences are summarized in Table 3. Except calcium ions, which did not accumulate in different concentrations according to distance from the emitter, all accumulations of all ions increased with increasing distance from the emitter. Potassium, chloride, and sodium ion accumulation was significantly greater at 60 and 90 cm in comparison to 30 cm from the emitter ($p$-value $< 0.001$ for all). Magnesium and nitrate also had significant increases in accumulation at 60 cm to 90 cm distance from the emitter ($p$-value $= 0.004$ and $0.001$, respectively).

**Table 3.** Summary of statistical analysis using Tukey HSD *p*-values, showing means for each parameter measured in each factor. Ions are shown as a total soil ion concentration. A single * in a factor indicates a significant difference of less than 0.05 between that value and others in its factor. ** indicates significant differences (*p*-value < 0.05) between all values in the factor.

| Parameter | Unit | Depth | | Irrigation | | Distance from Emitter | | | Time | | |
|---|---|---|---|---|---|---|---|---|---|---|---|
| | | 0–25 cm | 25–50 cm | 60% | 80% | 30 cm | 60 cm | 90 cm | Baseline | End of Year 1 | End of Year 3 |
| Electrical Conductivity | dS/m | 23.31 | 23.99 | 26.96 * | 20.34 | 18.66 * | 31.88 ** | 20.43 | 20.10 | 20.72 | 30.12 * |
| Sodium Absorption Ratio | mEq/L | 14.13 | 15.24 | 16.76 * | 14.14 | 9.78 | 18.02 * | 18.56 * | 10.89 * | 19.29 ** | 16.18 |
| Exchangeable Sodium Percentage | % | 15.28 | 16.59 | 17.92 * | 13.63 | 10.48 | 18.55 * | 18.53 * | 12.44 * | 20.43 ** | 16.78 |
| Magnesium | mg/L | 563.89 | 562.41 | 649.03 * | 485.71 | 474.96 * | 603.63 | 623.52 ** | 410.51 * | 588.47 | 703.13 ** |
| Calcium | mg/L | 5811.46 * | 4347.24 | 5229.56 | 5685.56 | 5192.70 | 5580.56 | 5599.42 | 1270.53 * | 7348.61 | 7753.55 ** |
| Potassium | mg/L | 559.81 | 502.37 | 535.69 | 598.27 * | 474.48 | 613.82 * | 612.65 * | 250.45 | 746.45 * | 704.04 * |
| Nitrogen (as Nitrate) | mg/L | 117.37 | 89.03 | 107.07 | 85.00 | 18.72 * | 100.99 | 168.38 ** | 190.65 ** | 60.18 | 37.26 * |
| Chloride | mg/L | 3055.71 | 3297.76 | 3912.15 * | 2348.89 | 1366.04 * | 4057.39 ** | 3968.14 | 4907.27 ** | 2899.47 | 1584.84 * |
| Sodium | mg/L | 1372.50 | 1412.74 | 1556.96 * | 1160.67 | 749.72 | 1645.67 * | 1681.05 * | 1870.86 ** | 1189.78 | 1015.81 * |

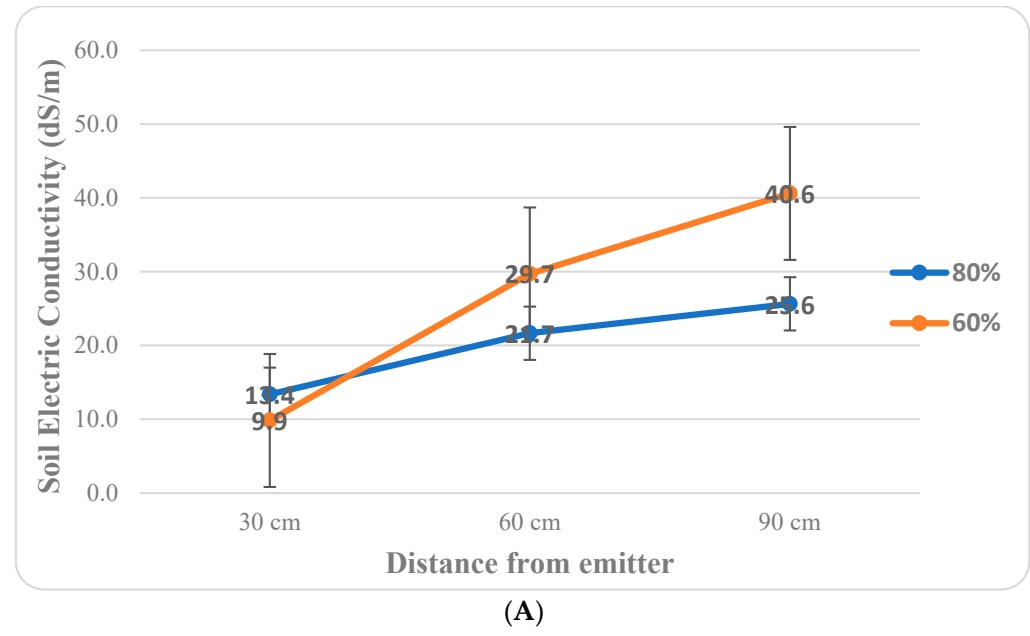

**(A)**

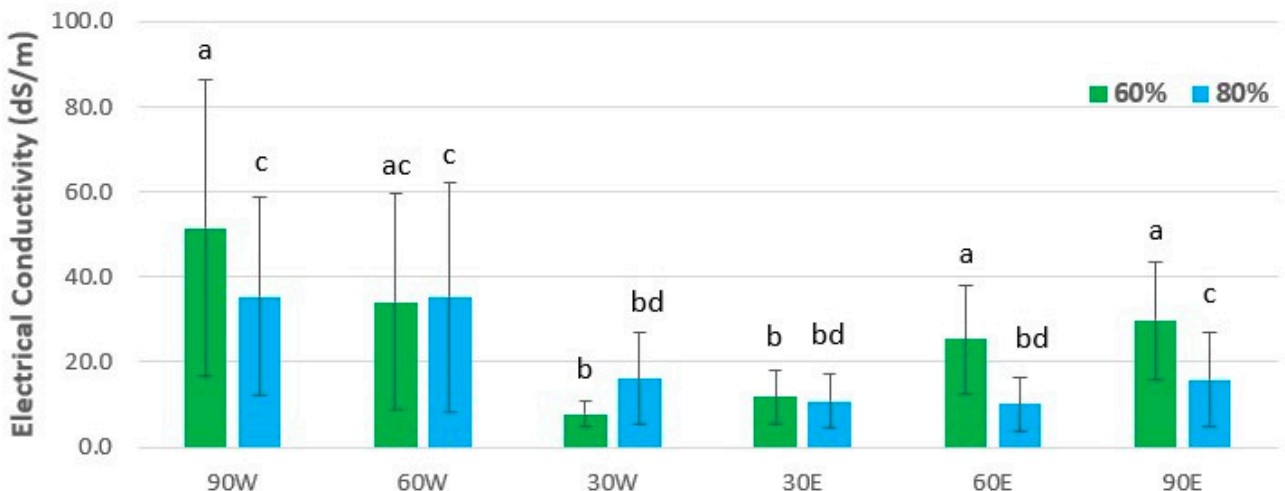

**(B)**

**Figure 1.** Mean electrical conductivity relationship between distance from the emitter and irrigation treatment. (**A**) The interaction plot shows a significant relationship between the distance from the emitter and irrigation levels in EC levels. (**B**) EC levels surrounding the irrigation emitter begin to show an ellipsoid pattern. Different letters (a through d) indicate a difference in significance, while error bars indicate standard error within the data.

### 3.4. Depth from Soil Surface

Calcium ion concentration significantly decreased from surface levels at 0–25 cm in depth to 25–50 cm in depth ($p$-value < 0.001). Large deposits of gypsum are located near the research site. While we do not have wind data, wind could potentially carry gypsum into and out of the research site, allowing for a greater accumulation of calcium on the surface, but lacking evidence of calcium accumulation below the surface. Other ions in the study did not show significant differences in accumulation by depth (magnesium $p$-value = 0.965; potassium $p$-value = 0.068; nitrate $p$-value = 0.2; chloride $p$-value = 0.764; sodium $p$-value = 0.767) (Table 3). Potassium and nitrate hand a tendency to decrease in accumulation from the surface to deeper depths.

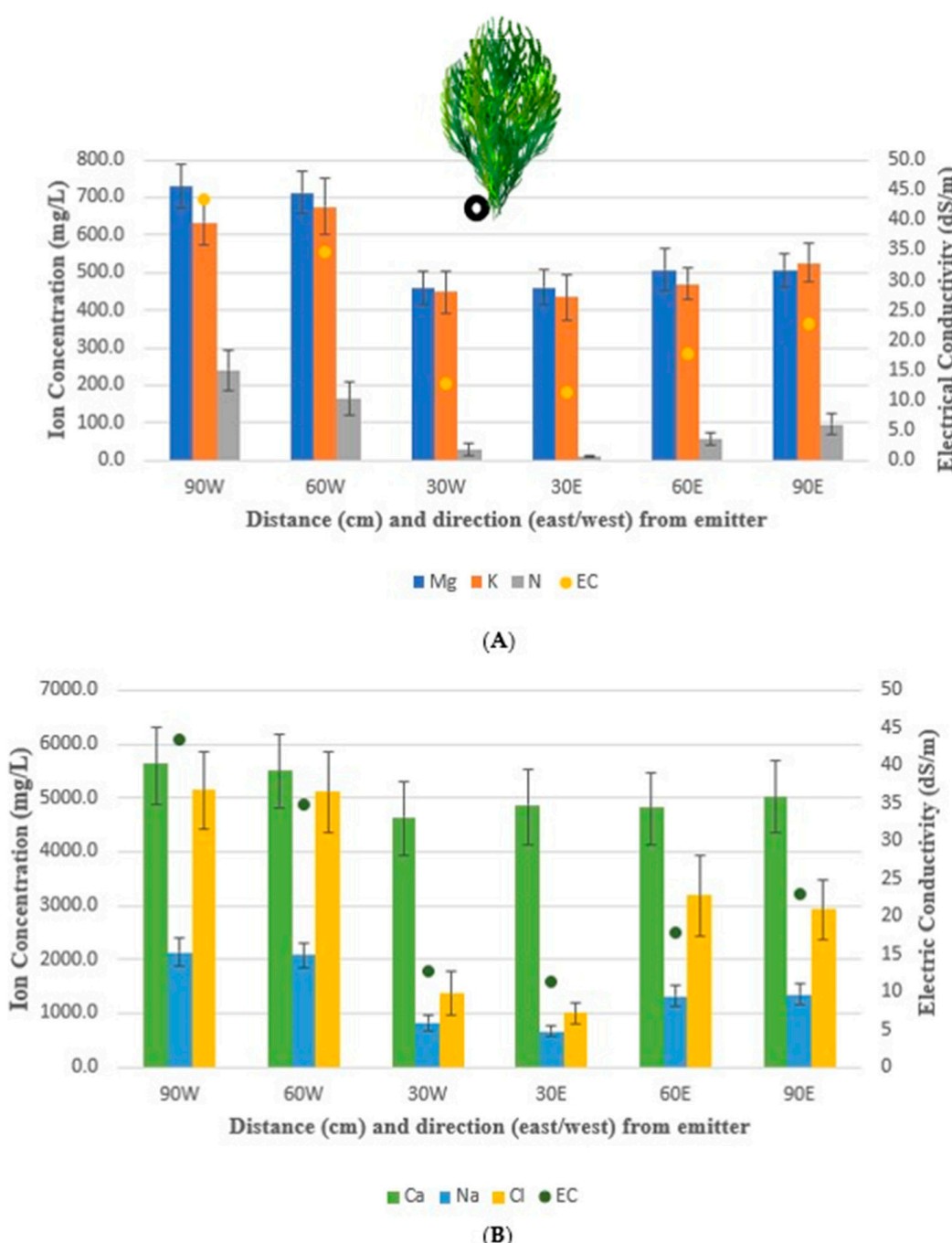

**Figure 2.** Mean ion concentrations separated by distance and direction from the irrigation emitter. (**A**) shows magnesium, potassium, and nitrogen ions; while (**B**) shows calcium, sodium, and chloride ions. Electrical conductivity (EC) is shown on both graphs, with secondary axis. Both (**A**) and (**B**) show the location of the plant and emitter for orientation of accumulation of ions. Statistical differences are not indicated, as this figure just shows the general patterns of ion accumulation, however, standard error bars are provided.

*3.5. Irrigation Treatments*

Magnesium, sodium, and chloride ions were in significantly higher concentrations in the 60% ET0 irrigation treatments than in the 80% ET0 irrigation treatments ($p$-value < 0.001, $p$-value < 0.001, and $p$-value < 0.001, respectively, for each ion) (Table 3). Under 80% ET0 irrigation treatment, water was applied for 32 more minutes in each irrigation than under 60% treatment, and this potentially allowed the ions to move and accumulate further away

from the tree trunk along the wetting front. Potassium ions accumulated in a significantly higher concentration in the 80% ET0 irrigation treatments (*p*-value < 0.001). Calcium ion accumulation did not demonstrate significant differences between irrigation treatments (*p* = 0.075). This could be because the soils of the region are rich in calcium.

### 3.6. Duration of Study

Over the three-year study period, magnesium, calcium, and potassium ions significantly increased in concentration from base-line samples collected in April 2017 to final samples collected in November 2019 (*p*-value < 0.001 for each ion), while nitrate, chloride, and sodium ions significantly decreased from baseline to the end of year three (*p*-value < 0.001, *p*-value = 0.003, and *p*-value < 0.001, respectively) (Table 3). We found 1270.53 mg/L of calcium ions and 250.45 mg/L of potassium ions at the beginning of the study in baseline samples. Potassium increased nearly three times to 704.04 mg/L by year three, while calcium increased six times to 7753.55 mg/L. Nitrogen decreased five times its initial measurement (from 190.65 mg/L to 37.26 mg/L) at the end of year 3.

### 3.7. Aboveground Plant Growth

Plant growth was not significantly decreased in the first year, but by the end of year three, as the soil salinity increased (as indicated by the EC), the aboveground growth decreased (*p*-value = 0.024). In the 80% ET0 irrigation, plants were larger (*p*-value = 0.013) than in the 60% ET0 irrigation treatment.

## 4. Discussion

As expected, the accumulation of ions along the wetting front of the irrigation took on an inverted bell pattern for most of the ions studied, although the pattern varied for some ions. In accounting for the direction of the distances for electrical conductivity, we found large EC values at 90 cm (west) of the emitter, supporting the inverted bell curve pattern. However, statistically, 60 cm from the emitter had significantly higher EC values at 31.88 dS/m. Additional sampling at smaller intervals and deeper depths could provide more evidence with which to determine the pattern shape of accumulation. It is believed that the wetting front in the soil allowed the ions to accumulate near the surface at 60 cm distance from the emitter but dropped below 50 cm depth before the 90 cm distance, therefore distorting the pattern. Due to the high solubility of sodium, soil salinity is mostly composed of sodium chloride; however, salts made from calcium, magnesium, and potassium are common in arid and semi-arid soils [38]. Soil salinity increased in our study area over a three-year period from an average EC of 20.10 dS/m to 30.12 dS/m due to the addition of brackish irrigation water. We added approximately 1.5 kg of sodium, 1.4 kg of calcium, 1.8 kg of chloride, 0.7 kg of magnesium, 13.5 g of nitrogen (as nitrate), and 7.9 g of potassium ions to the soil over the course of our study. The final destination of the ions was not explicitly studied as some ions would have been taken up by the roots of the plants while some ions would have moved below the study range in the soil. While sodium and chloride were large contributors to the salinity of the irrigation water, calcium was present in similar concentrations as well.

The SAR value of our soils significantly increased from the baseline to the end of year one and then decreased by the end of year three. The sodic soils reduce water flow through the soil [13,14] as the porosity is changed. Our study plots were not manipulated using a plow, and no machinery was used in the site. Taking core samples throughout the study to examine porosity changes would have been beneficial. Cucci et al. [39] found that while changes in pore size and distribution did occur when irrigating with sodic saline water, there was no permanent macroscopic change, and the pore size changes were recovered when the soil was treated with soil conditioner. Soil conditioners tend to be high in calcium content. It has been shown thar the addition of sodium to soil has a negative effect on hydraulic conductivity, especially in clay soils [14,39–41]. Hydraulic conductivity was not considered for this study. Rasouli et al. [40] found that the addition of gypsum to

agricultural plots decreased the SAR and pH of the soil, while increasing soil salinity and hydraulic conductivity. While we did not add gypsum intentionally to the study soil, the high calcium content of the groundwater used, as well as the addition of natural gypsum through wind events from nearby White Sands National Park, may have contributed to the data we collected. Exchangeable sodium was not distinguished from total sodium in our study. We assume that the high solubility of sodium allowed for sodium ions to be present in soil pore water. We calculated ESP from the SAR values in our soil. In our calculations, the ESP was approximately 12%; it increased to over 20% by the end of year one and decreased to just below 17% by the end of year three. While ESP was decreasing, we still observed a reduction in plant growth, which could be attributed to the ESP being greater than 15% [35,41,42].

It has been suggested that there is an exchange of monovalent ions and divalent ions in clay soils [42]. We saw a decrease in sodium ions in our soil samples at the end of year three in comparison to the baseline. While sodium exchange may be occurring, the calcium addition from the water source is negating the build-up of sodium. Sodium/calcium ratios decreased significantly from 1.49 to 0.15 from baseline to EOY 3 ($p$-value < 0.001), while calcium/magnesium ratios increased significantly from 3.55 to 12.4 in the same period ($p$-value < 0.001).

Both potassium and nitrate, along with magnesium, are necessary in plant enzymatic activity, and potassium plays a large function in photosynthesis [43]. Plant roots may be taking up these ions at depths greater than 25 cm, resulting in a reduction in the accumulation of these ions at deeper depths. Soil sampling below 2 m was recommended in peach orchards [44] to ensure that plant roots were not affecting the accumulation of salts. In additional studies [45], water and nutrient movement has been shown to be variable up to 1 m below drip irrigation sources in vineyards. Future studies with *Atriplex* should aim to collect soil samples at 2 m or deeper to track salt accumulation with depth and to determine if the root uptake or leaching of ions is occurring below the rootzone.

It should be noted we do not know all the ions which *Atriplex* species uptake in their soil remediation process, or the metabolism of the ions, but it is suggested that potassium, magnesium, sodium, and zinc ions are accumulated in some *Atriplex* species' leaves [22]. Our earlier study [46] documented higher calcium and magnesium concentrations in *Atriplex canescens*, as well as lower osmotic potentials. However, we did notice that *A. canescens* excreted ions include sodium on leaf surfaces [22,47]. The uptake of sodium and chloride ions has been documented in *Atriplex hamilis* [48]. In our study, there is a decrease in potassium and sodium closest to the root zone of the plants, but it cannot be determined if this was due to an increase in uptake by the plant or if the accumulation was prevented due to the ions moving away from the rootzone during irrigation.

While calcium levels increased significantly between the baseline samples and end of year three samples ($p$-value < 0.001), the accumulation of calcium ions decreased with distance from the plant. After chloride, calcium ions were of the greatest quantities in the water source (Table 2) of the ions we studied. The increase in accumulation of calcium ions in the soil was expected. While other ions may also be attracted to the negatively charged clay particles, calcium is accumulating at a greater rate than other ions introduced to the soil. The negative charge of chloride does not permit the ion to attract soil particles. However, the accumulation levels of chloride ions in the soil suggests that the negatively charged ions are attracted to other ions or to material in the soil.

The end-of-year-one soil samples at 30 cm (both depths) had ECs within the range of 8–16 dS/m, which salt-tolerant plants can withstand [36]. Further from the plant base, the soil is above 16 dS/m, which very few plants can grow in such conditions [36]. Only one set of samples (those closest to the emitter) was within the salt-tolerant range at the end of year 3, as described by Lamond and Whitney [36]; however, *A. canescens* and, to a lesser extent, *A. lentiformis* still showed growth aboveground. Salt accumulation was observed in all samples by the end of the study. To reduce salt accumulation in soil, leaching is commonly used to move salts down and away from the root zone [49]. Flood irrigation

is the traditional method, in combination with drainage, of relieving salt accumulation. Li and Kang [49] showed that with irrigation, the water table becomes shallow, which increases the soil salinity. Li and Ren [50] reported model- simulated leaching of 80% at 450 mm precipitation in loam, sandy loam, and clay loam experimental sites. Our summers consisted of rainfall equaling an average of approximately 185 mm, with 205 mm in 2018 from May to October being the greatest period of rainfall. Accumulation of salts is expected due to the lower precipitation available to leach salts away from plant roots. The plants in the field plots are 5 years old at the time of the year-3 sampling. Root development for the study plots was sufficient for continued growth as soil salinity increased. Further research is needed in order to determine the threshold of accumulation of soil salts which would affect *Atriplex* plants. The growth stage of *Atriplex* may also be affected differently at different salinity levels. Additional studies to determine soil salinity after irrigation has concluded are being considered in order to determine if precipitation and phytoremediation continue to remove accumulated salts.

Over the three-year study, the *Atriplex* declined in aboveground growth. Contrary to Flores et al. [46], soil salinity in this study raged between 7.4 and 123.1 dS/m (with the average being 29.7 dS/m) and may have contributed to the decrease in aboveground growth. However, factors related to weather and management, including trimming, could have contributed to the decrease in aboveground biomass. Currently, there are no trimming or harvesting guidelines for *Atriplex*. Research [51] has shown *Atriplex* species to have an increase in woody stems with a reduction in leaf tissue over successive cuttings, which could explain the decrease in growth we observed in our study. Cultivation practices still need to be developed in order to ensure the best growth conditions for these plants.

Further studies on ion accumulation in soils, with a focus on which ions the *Atriplex* plants are absorbing, are needed in order to be able to determine if either species is appropriate for growth for cattle fodder. Ventura et al. [52] discuss *Atriplex* as a potential fodder halophyte but recommend that the plant be mixed with other halophytic species to reduce salt intake to animals. As a stand-alone feed, the nutritional value may contain too much sodium for animal ingestion [52].

## 5. Conclusions

Drip irrigation is recommended in arid and semi-arid regions in order to reduce evapotranspiration. As soil salinity increases in these regions around the world, it is necessary to consider how RO-concentrated waste used as an irrigation source for halophytic plants might be affecting the environment. We found that the average EC of the soil increased as we added brackish water to simulate concentrated waste. The average SAR increased throughout the study when the irrigation levels were 60% ET0. After an initial increase in the first growing season, the average SAR decreased by the end of year 3 to under 80% ET0. EC patterns appear as an inverted bell under the drip irrigation emitter, but ion accumulation patterns were varied. Given that soils were not sodic, highly salt-tolerant halophytes can grow in the soil. This leads to the potential, if the groundwater table is deep, of using RO-concentrated waste to irrigate fodder plants, such as *A. canescens* or *A. lentiformis*, on a large scale without causing damage to the environment. Occasional leaching of the soil is recommended in order to reduce ion accumulation during times of lower evapotranspiration to safeguard soil properties. This will also encourage better yields of aboveground *Atriplex* as plant growth decreased through the duration of our study. Further studies are needed in order to identify the salinity thresholds and harvesting guidelines for *Atriplex* before moving towards marketing the plant.

**Supplementary Materials:** The following supporting information can be downloaded at: https://www.mdpi.com/article/10.3390/soilsystems7040084/s1. Figure S1: Weather data through study, Figure S2: Study site information, Table S1: Soil Texture Data, Figure S3: Electrical conductivity in comparison to ion concentration, Figure S4: SAR interaction plot.

**Author Contributions:** Each author has contributed to the production of this manuscript and has approved the submission of the article to be published. S.M.C. assisted in funding proposal and research design, completed the research activities, analyzed the data, and conducted original draft preparation and revisions. M.K.S. assisted in supervision, funding acquisition, research design, data analysis, and production of the paper, including significant written review. S.J. assisted in the statistical analysis of data and the review of the paper. S.O. participated in funding acquisition and the editing of the manuscript. All authors have read and agreed to the published version of the manuscript.

**Funding:** This research was funded by the Bureau of Reclamation [Grant number ST-2021-1780-001-05], with support from the NMSU (New Mexico State University) Agricultural Experiment Station, the National Institute of Food and Agriculture and the Nakayama Professorship.

**Institutional Review Board Statement:** Not applicable.

**Informed Consent Statement:** Not applicable.

**Data Availability Statement:** Data available at Reclamation Information Sharing Environment (https://data.usbr.gov/catalog/4623/item/11468).

**Acknowledgments:** We appreciate the help of Randy Shaw, facility director of the Brackish Groundwater National Desalination Research Facility in Alamogordo, New Mexico, with his staff, for the upkeep of the *Atriplex* study plots.

**Conflicts of Interest:** The authors declare no conflict of interest.

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
