# Peer review of "The Effects of Brackish Irrigation on Soil Ion Accumulation and Growth of Atriplex Species"

_soilsystems, doi:10.3390/soilsystems7040084_

Round 1

Reviewer 1 Report

The paper titled “Continuous Drip Irrigation with Brackish Water: Changes in Soil Characteristics and Growth of Atriplex Canescens and A. Lentiformis” by Cerra and coathors was to research changes in the characteristics of Soil Characteristics and Growth of two plants with Irrigation. The paper is original and interesting. The results of paper will provide some instructive meanings for evaluating the management in agricultural. However, current version of paper need to further improved prior to publishing. Please see the comments below.

1) The title is too long, and it is not specific and explicit, should express the key content in your research.

2) The abstract is also too long and the focus is not clear. You should state it in the following order: background, purpose, method, result, conclusion, and scientific significance.

3) Line 12: what is “ET0”? Please explain.

4) Introduction, a hypothesis is very necessary.

5) Materials and Methods, I suggest inserting a table detailing the parameters of your experiment.

Figures 1 to fig 3 can be combined into a single graph.

6) The discussion should be strengthened further in this current version, arrounding the critical results and highlights in this study and then compared the results with previous articles in elsewhere.

Minor editing of English language required

Reviewer 2 Report

The studies carried out by the authors are relevant and practical. The article may be published after some little revision.

It is important that the manuscript is organized along editorial lines. Eliminate/replace patents

Additional comments:

I have noticed in the manuscript both problems of a stylistic nature and lack of detailed information in the results paragraph and even more so in the discussion section.

The language needs some refinement. 

Reviewer 3 Report

In general, the manuscript by Cerra et al. is good contribution to the soil systems as it describes some measures for the reclamation or management of saline soils under continuous use of brackish water. The manuscript is well written, however, it needs minor revisions before it gets accepted by Soil Systems. The comments are as below:

1. Title is inappropriate. Please revise.

2. Better to write electrical conductivity rather electric conductivity. Also follow the same suggestion throughout the manuscript.

3. In abstract, please write concentrated waste instead of concentrate waste. Same suggestion for the whole manuscript.

4. Your hypothesis is that above-ground parts of plants are not affected by distant accumulation of concentrated waste from the source but your results are contradictory to your hypothesis. Please revise your discussion to justify your contradiction.

5. GIS map would be better for description of sampling points/locations.

6. It is not clear that how many soil and water samples were collected. In addition, please make it clear that how many replications of each treatment were established for your experiment.

7. In table 1, pH of the water samples is 7.35 despite all the parameters of salinity are high. Please verify the results of pH. Why authors did not reported Na concentration in the table 2?

8. What is SAR? It must be explained for the first time used in the manuscript.

9. Please check the results of Figure 4B. These are not-significant at both 60 and 80% ET0 levels.

10. in table 2, write 0-25 instead of O-25. Its typo error.

Minor English editing is required.

Round 2

Reviewer 1 Report

Thank you very much for making extra efforts to revise this article. But in my opinion, there are still several problems that need to be revised:

(1) The abstract lacks conclusion and meaning.

(2) The hypothesis should be presented according to the research objectives

(3) The presentation of the figure is too rough, and a more reasonable typesetting is needed, for example, many figures can be merged.
